# Academic performance of children in relation to gender, parenting styles, and socioeconomic status: What attributes are important

**Nayab Ali** [1] *, **Asad Ullah**[2], **Abdul Majid Khan**[3], **Yunas Khan**[4], **Sajid Ali**[5], **Aisha Khan**[6], **Bakhtawar**[7], **Asad Khan**[8], **Maaz Ud Din**[9], **Rahat Ullah**[10], **Umar Niaz Khan**[10], **Tariq Aziz**[1], **Mushtaq Ahmad**[3]

1 Department of Sociology, Kohat University of Science and Technology, Kohat, Khyber Pakhtunkhwa, Pakistan, 2 Department of Rural Sociology, The University of Agriculture, Peshawar, Khyber Pakhtunkhwa, Pakistan, 3 Department of Sociology & Psychology, University of Swabi, Swabi, Khyber Pakhtunkhwa, Pakistan, 4 Head of Department of Pakistan Studies, Islamia College, Peshawar, Khyber Pakhtunkhwa, Pakistan, 5 Department of Social Work, Kohat University of Science and Technology, Kohat, Khyber Pakhtunkhwa, Pakistan, 6 Department of Sociology, The Women University, Multan, Punjab, Pakistan, 7 University of Malakand Women Campus, Khyber Pakhtunkhwa, Khyber Pakhtunkhwa, Pakistan, 8 Department of Tourism & Hotel Management, University of Swabi, Swabi, Khyber Pakhtunkhwa, Pakistan, 9 Department of Business Administration, ILMA University, Karachi, Pakistan, 10 Department of Law, Kohat University of Science and Technology, Kohat, Khyber Pakhtunkhwa, Pakistan

* nayabaup@gmail.com

**Data Availability Statement:** All relevant data are within the manuscript.

**Funding:** The author(s) received no specific funding for this work.

## Abstract

What are the effects of parenting styles on academic performance and how unequal are these effects on secondary school students from different gender and socioeconomic status families constitute the theme of this paper. A cross-sectional and purposive sampling technique was adopted to gather information from a sample of 448 students on a Likert scale. Chi-square, Kendall's Tau-c tests and hierarchical multiple regression analyses were used to determine the extent of the relationship among the variables. Chi-square and Kendall's Tau-c ($T^c$) test results established that the socioeconomic status of the respondent's family explained variation in children's academic performance due to parenting style; however, no significant difference was observed in the academic performance of students based on gender. Furthermore, hierarchal multiple regression analysis established that the family's socioeconomic status, authoritative parenting, permissive parenting, the interaction of socioeconomic status and authoritative parenting, and the interaction of socioeconomic status and permissive parenting were significant predictors ($P<0.05$) of students' academic performance. These predictor variables explained 59.3 percent variation in the academic performance of children ($R2 = 0.593$). Results of hierarchal multiple regression analysis in this study ranked ordered the most significant predictors of the academic performance of children in the following order. Family socioeconomic status alone was the strongest predictor ($\beta = 18.25$), interaction of socioeconomic status and authoritative parenting was the second important predictor ($\beta = 14.18$), authoritative parenting alone was third in importance ($\beta = 13.38$), the interaction of socioeconomic status and permissive parenting stood at fourth place in importance ($\beta = 11.46$), and permissive parenting was fifth ($\beta = 9.2$) in influencing

**Competing interests:** The authors have declared that no competing interests exist.

academic performance of children in the study area. Children who experienced authoritative parenting and were from higher socioeconomic status families perform better as compared to children who experienced authoritarian and permissive parenting and were from low socioeconomic status families.

## Introduction

Academic performance is the process of acquiring knowledge and skills that provide the foundation for development [1]. Academic performance refers to the "achievement of educational benchmarks", therefore, most of the educational efforts of students, teachers, parents and educational institutions revolve around achieving these educational goals to provide a sound foundation for national development and meet the challenges of the modern world [2–4]. Measurement of academic performance via standardized tests and examinations and its verification in a system of grade marks or percentage points is a universally accepted norm [5, 6]. Good academic performance has been related to successful development trajectories and better life performance for the persons possessing it as well as for national development. On the contrary, poor academic behavior is linked to academic failure, maladjustment, poor access to services and opportunities or alternatively access to low-paying and less rewarding jobs and low productivity in later life. Also, when a state has a larger population of people with inferior academic credentials, it is less able to implement productivity-boosting technology and innovative working methods, which ultimately causes the nation to fall in the international rankings for its socioeconomic standing [7]. The reasons for the poor academic achievements of students in poor and developing countries are almost similar. Several factors are associated with the academic performance of students. These factors include individual, social, economic and institutional. Some of the most important predictors of child academic achievements include school attendance, student's interest in the study, hard work, dedication, self-confidence, family support, parenting style, family socioeconomic status, school environment and neighborhood facilities [8–10], peer influence and parent involvement in children education [11, 12]. Similarly, Radhika found that classroom size, teaching methodology, teachers' capabilities, facilities and learning environment at school affect the academic performance of students [13].

This study is designed in light of Bronfenbrenner's socio-ecological model. Bronfenbrenner's socio-ecological framework provides a theoretical foundation to connect multi-layered patterns of interaction of personal relationships and social settings in institutions that shape students' behaviour, learning motivation and academic performance. Thus, the academic performance of children from the study area is supposedly having links with some micro-, meso-, exo- and macro-level systems that are explained under Bronfenbrenner socio-ecological model. However, due to time and financial constraints, as well as the direct influence of the two levels (Micro and Meso levels) of Bronfenbrenner's socio-ecological model, the current study focused only on these two levels. This research study has three main objectives: (1) to examine the association between parenting style and academic performance of children, (2) to know about the variation in academic performance of children with respect to parenting style on the basis of student gender and family socioeconomic status, and (3) to measure the relationship between parenting style and family socioeconomic status on the academic performance in isolation and interaction with each other.

### National and international scenario

According to the Global Human Capital ranking of 130 countries, the less-developed and developing countries like African and South Asian countries lack behind the developed

countries and make up the lower end of the regional rankings due to poor investment in education, low-skilled workforce, poor utilization of skills and little know-how of utilization of skills. Moreover, out of 214 million children in Sub-Saharan Africa and the Central and Southern Asia states do not achieve minimum proficiency levels in reading and mathematics and 81% of these children have ages of 6 to 14 years [14]. According to the Global Human Capital ranking of 130 countries, Pakistan stands at 125th, which is well below the neighboring countries in South Asia such as Sri Lanka, Nepal, Bangladesh and India which ranks 70, 98, 111 and 103 respectively [15]. Pakistan, the 5th most populous country in the world, has a population of more than 200 million. More than two third of its population live in rural areas. Between urban and rural populations, there are substantial disparities. For instance, the literacy rate is 53% in rural areas whereas 76% in urban ones. Urban areas (18% rural compared to 74% urban) have approximately four times the likelihood of having access to vital developmental services [16]. According to UNICEF in addition to the 40 million that are enrolled in school, there are presently 22.8 million more Pakistani children not in school. Following Nigeria, this country has the second-highest proportion of out-of-school children worldwide. 5.3 million of the 22.8 million children in this group are dropouts, and 17.5 million have never attended school [17]. The average percentage of students who passed all subjects in the science and arts groups in 2019 was 68.24% and 36.83% respectively. In 2018, the scientific group had a cumulative average pass percentage of 54.99%, while the arts group had a pass percentage of 38.99%. In 2017, the science group had a cumulative average pass percentage of 60.05%, while the arts group had a pass percentage of 42.29%. In 2016, the science group's cumulative average pass rate was 54.11%, while the arts group was 44.95%. In 2015, the science group's cumulative average pass rate was 46.92%, while the arts group was 34.22%. In the same way, a high number of students took the exams in 2018, 2017, 2016, and 2015, but fewer students passed with low marks, which was an alarming situation in secondary education in Pakistan [18, 19]. On a regional basis in the country, District Malakand is on top as per the last report of Alif Ailaan and SDPI on education scores. However, the last result of passing students in the Secondary Schools Certificate (SSC) examination is 69.73 percent and 30.27 percent of students remained unsuccessful [20].

## Theoretical background

From long ago educationists and researchers are interested in exploring various determinants of students' academic performance ranging from child personal factors to contextual factors. Some of the most echoed theories are achievement goal theory, self-determination theory, social learning theory, and socio-ecological model. The achievement goal theory explains that students' academic achievements are associated with students' personal factors i.e. their personal goal orientations and self-determination. The theory bifurcates the personal goal into two main categories namely mastery goal (goals for personal improvement and gaining knowledge for one's own sake) and performance goals (to perform to supersede others in terms of academic output). These two types of goals help children in their development and learning (understanding and performing hard tasks) and keep them from failing through performance (to outperform and beat others) respectively [21–24]. Empirical research based on achievement goal theory made it evident that mastery goal improved students' interest in learning, and enhanced their self-efficacy, self-determination and cognitive skills, whereas performance goal focused on the exercise of abilities for success [25–27]. A combination of mastery and performance goals contributes towards overall academic achievements [28]. The achievement goal theory was criticized for its broader applicability as its main focuses are on interpersonal predictors of achievements at individuals' personal or at a very micro level [29]. Other

researchers found some conflicting outcomes that were associated with performance goal orientations. Furthermore, some of the students, despite their maladaptive behavior towards performance and goal orientations, secured better grades which are inconsistent with the achievement goal literature [30]. Moreover, this theory could not take into consideration multiple dimensions at broader levels to explain the academic performance of children holistically [31].

Self-determination theory (SDT) is another theory to explain students' classroom performance and is based on the premise of motivation. The theory assumes that all students possess innate tendencies for growth (intrinsic motivation) which is the motivational foundation for better classroom engagement and appropriate school functioning [32–37]. The theory further explains that motivation from teachers and enabling facilities at school (extrinsic motivation) provide a further boost to the inner motivational resources of students in facilitating their high-quality engagement, therefore, the academic achievements of students are reliant on intrinsic and extrinsic motivations [38]. This motivational phenomenon presented and explained self-determination theory along with its associated five other theories i.e. basic needs theory, organismic integration theory, goal contents theory, cognitive evaluation theory and causality orientations theory.

The basic needs theory focuses on psychological needs (autonomy, competence, and relatedness) and magnifies the importance of intrinsic motivation, positive engagement, effective functioning and psychological well-being [39]. Organismic integration theory explains extrinsic motivation and its relationship with students' academic socialization [40, 41]. Goal contents theory compares intrinsic goals and extrinsic goals to elucidate how intrinsic goals support psychological well-being, whereas, extrinsic goals poster psychological ill-being [42, 43]. Cognitive evaluation theory is another micro theory that emerged from SDT that was developed to predict the positive and negative effects of extrinsic goals on intrinsic motivation Causality orientation theory is the fifth offshoot of SDT that identifies individual differences among students in terms of their motivation and engagement. It also reflects on the fact that some students prefer autonomy, whereas others perform better in a controlled environment [44].

The SDT theory, however, is criticized on its conceptual grounds as empirical studies have revealed some adverse effects of rewards on motivation. Moreover, the application of this theory to real life is considered doubtful due to complex social events. Furthermore, the motivation that is generated by rewards, in the context of complex tasks that make up most of the human lives in their profession, is short-term and shallow. Excessive focus on rewards is also found detrimental to creativity and true engagement. The critique of SDT, therefore, rightly says that the creation of an atmosphere in which people feel free to act independently and creatively towards shared goals is much harder. Other social psychologists say that self-determination theory (SDT) is still under development and searching for new avenues in social psychology to explore [45].

Social learning theory is based on Banduras' Bobo Doll experiments during the 1960s. This theory tries to establish that social learning is the outcome of observing and interacting with others. The theory was later on named "social cognitive theory" according to which learning occurs when there is reciprocal interaction of person between environments which results in a specific behavior. The interaction of a person with the environment stimulates behavior which in turn affects the person and the environment. Therefore, the learning process is a complex interplay of these factors which is termed by Banduras' as reciprocal determinism [46, 47]. Therefore, it is not only the students' belief in their abilities that shape their academic achievements, rather, the social environment at family, school, neighborhood, peers and mass media are also important in shaping the learning outcomes of children [48].

The social learning theory is criticized for disregarding the emotional or motivational basis of learning behavior. Moreover, the operationalization of this theory in its entirety is also

questioned. Some of the assumptions of the theory are disproved through empirical research for example the assumption that "the environment will bring changes in the person automatically", does not always stand true. In addition, the extent to which the factors of person, environment and behavior influence the actual learning behavior is not clear. The theory has also overlooked the biological determinism and maturation effect on the learning process [49].

After discussing the major theme and criticisms on the above theories, there emerges a desire to focus on such a model that can help understand the multiple layers of youth ecology that promote academic growth and limit negative educational outcomes in the children.

One of the most widely used models in this regard is Bronfenbrenner's socio-ecological model which explains various factors of child development at different levels [50–53]. This model visualizes the environment of each individual residing in society into individually observable distinct patterns of layers. The multi-layered pattern of interaction of personal relationships, social settings and institutions shapes students' behaviour, learning motivation and academic performance [54–56]. This theory divides these patterns of interaction into a system of four levels i.e. micro-level systems, meso-level systems, exo-level systems and macro-level systems. The micro-level system is the closest environment in which a student lives and interacts; it includes home, neighbourhood, and school. The meso-level system includes the interaction of two micro-level systems such as the interaction between parents and teachers and between parents and neighbours. The micro- and meso-level systems have a direct influence on child learning and performance. The other two levels, exo-level (parents, workplace and their association) and macro-level (culture, policies) system have no direct effect on the children and the children are not directly involved in that [57–59].

Empirical studies have also identified multiple determinants of children's academic performance that are systematically framed in Bronfenbrenner's socio-ecological model [60–62]. However, parenting style (a micro-level factor refers to specific behaviours and strategies used by parents to control, socialize and establish an emotional relationship with their children) and family socioeconomic standing seems to be the most emphasized factors [63–66]. Numerous studies show overwhelming evidence of the important role played by parenting style in influencing the academic performance of children [67, 68]. Based on various dimensions and characteristics of parenting, Baumrind identified three types of parenting styles that had profound effects on behaviour of children. The typology of parenting style included authoritative, authoritarian and permissive parenting. This typology of parenting style is based on responsiveness (warmth, clarity of communication, acceptance and involvement) and demandingness (control, supervision and maturity demands) as valued by parents [69]. Authoritative parenting is linked with a high level of both responsiveness and positive demandingness, the authoritarian parenting style is characterized by low responsiveness and high demandingness, and the permissive parenting style is based on high responsiveness and low demandingness [70]. Specific parental behaviour that influences the academic performance of children via authoritative parenting includes warmth and a democratic environment in the family, under which children openly participate in decision-making, discuss their concerns and have a positive relationship with their parents. From the parental side, authoritative parents enforce rules and norms, provide guidance and impose sanctions when necessary. Authoritarian parents are strict to the extent of harshness that limits child's participation in family discussions and sharing problems with parents. On the other extreme, permissive parents provide limitless freedom to the children and ignore their deviance with no or very low guidance to discipline the children [71].

Parenting style is an important predictor of the academic performance of children to the extent that in some cases it may even offset the negative effects of socioeconomic status and poor neighbourhood in achieving better academic grades. However, Sirin reported based on a

meta-analysis of 58 studies that parental socioeconomic status (education, occupation and income) is the strongest predictor of academic achievements and that low socioeconomic status leads to lower academic achievements of students [72, 73]. For some researchers, a secured socioeconomic status is at par with other micro-level variables such as parenting style, home, school and neighbourhood. in shaping the academic achievements of the children, and for others, the effect of socioeconomic status was merely like a catalyst that boosted the academic performance of children when combined with appropriate parenting, positive peers, secure living and conducive school environment. The home locality, access to health and access to educational and recreational services are the functions of high socioeconomic status. Whereas, low enrolment and low academic performance are common in families with low socioeconomic status [74, 75]. Gender is another dividing line that distinguishes the academic performance of male and female children. In egalitarian societies, the gender-based distinction is not as wider as compared to that in a patriarchal society where males are preferred over females [76, 77]. Furthermore, some aspects of academic learning like engagement with school, self-esteem and enthusiasm for studies were excelled more by girls than boys, however, the socioeconomic system in patriarchal families was more favourable to boys than girls to support them in achieving higher academic grades [78].

## Literature review

Biological determinists lay huge emphasis on hereditary and biological factors in understanding behaviour-related problems in children. At the same time, certain social factors may also prove deterministic in putting youth in a disadvantaged position in an unequal society, as the benefits of some interventions may yield different consequences for youth from different socioeconomic groups. Therefore, the stories of psychological stresses, mental illnesses, academic failures and unsuccessful life are more pronounced in the poor segment of society [79]. A review of international and national literature discloses a gradual decrease in school dropout rates during past decades. However, a major chunk of educated folk did not acquire appropriate academic grades. As a result, most of them remain unemployed because they barely manage to enter the labour force [80, 81].

The global picture of human capital ranking shows that the poor and developing countries of South Asia and Africa including Pakistan are placed near the bottom of this ranking due to poor investment in education, low-skilled workforce, poor utilization of skills and little knowhow of utilization of skills. The government in Central and Southern Asia and Sub-Saharan Africa invested heavily in the education sector to achieve the Millennium Development Goal of universal primary education for all children by 2015. This investment resulted in an increase in the net enrolment rate. However, these countries are still faced with the challenges of retention rate and quality education [82]. World Bank carried out a survey and reported that worldwide 617 million children and adolescents are not been able to achieve minimum proficiency levels in reading and mathematics. Among these children, next to Sub-Saharan Africa are the Central and Southern Asia states where 81% of children and adolescents are not achieving proficiency outcomes [83]. The literacy percentage in Pakistan, at 57%, is significantly lower than that of its neighbouring countries. Given that primary school is where formative learning occurs, the dropout rate of 22.7 Percent (third highest in the region after Bangladesh and Nepal) is a grave issue. According to the 2016 Global Education Monitoring Report of the United Nations, Pakistan is 60 years behind in secondary education and 50 years behind in basic education in terms of meeting international educational standards. The children not attending school at the elementary, secondary, and upper secondary are 5.6, 5.5, and 10.4 million in numbers respectively. This results in an alarming and mind-boggling situation for the whole nation [84, 85].

It has been highlighted by a meta-analysis that the reasons for poor educational outcomes in developing and poor countries are almost the same. It has been observed that poor educational outcomes in South Africa are attributable to certain factors ranging from students' personal factors to school and home-related factors [86]. For instance, home-related factors include a lack of parental support towards child education, a non-conducive learning environment at home and poor socioeconomic status of parents. All these home factors lead to poor academic performance in children [87]. A study carried out by Jekonia in Finland concluded that school performance has a positive correlation with an authoritative parenting style. While, it was discovered that children's academic achievement and authoritarian and permissive had a negative correlation [88]. The author assessed the academic performance of 345 students with respect to parenting styles in Lebanon and found that adolescents who believe their parents as authoritative are more likely to form strong efficacy convictions and greater intention, and as a result, they are more likely to perform well in school than their peers with authoritarian and negligent parents [89]. However, the study also revealed that the impact of parenting on academic success was not moderated by socio-demographic factors. Similarly, a study conducted a study in Sirjan, Iran, the sample comprised 82 high school students, including 251 females and 131 males and concluded that having a supportive parenting style was very significant for children's educational success and goal orientation [90]. A research study in Indonesia assessed the impact of parenting style, age and size of the family on students' academic performance. It was illustrated that at the adolescence stage, the students spend most of their time in extra-curricular activities paying little attention to their studies and consequently their parents have to adopt an authoritarian parenting style. Moreover, in a family of large size, the parents are likely to adopt strictness and punitive techniques for socializing their children which results in poor academic achievements [91]. According to a research study by Farid et al. in Pakistan, parental practices have a significant influence on teens' personalities. This study found a strong correlation between parental responsiveness and teenage personality development. Adolescents' personality development is also impacted by parental influence over them. Adolescents' personalities will develop attributes of confidence if their parents adopt an authoritative parenting style to properly manage their children's expectations and attentiveness [92]. A wide range of studies established that, despite variations from culture to culture, a balance of responsiveness and positive demandingness in parenting style is related to higher academic performance [93–97]. The authors declared that the achievements of the student are associated with tangible and non-tangible resources both at school and home. Therefore, parenting style is one of the non-tangible assets that influenced the academic achievements of the students [98]. Similarly, in a meta-analysis by Bernstein, several parenting styles being important in affecting the academic achievement of students were compared. Authoritative parenting style, despite its variation from culture to culture, is a strong predictor of higher academic achievement in children [99]. Despite dilution of the positive effects of authoritative parenting style in autocratic society (such as in Asia and Africa), such parenting style shows consistently promising results in terms of the academic achievements of children. The effect of authoritarian parenting on positive educational outcomes in children was somewhat more promising in autocratic cultures (Asia and Africa) than in democratic cultures (America and Europe). However, the results of permissive parenting in this regard remained at the bottom in most cultures. Pinquart and Kauser in their analysis of 428 published studies, throughout the world, in 2017 and concluded that the authoritative parenting style was associated with positive academic outcomes as compared to authoritarian and permissive parenting, therefore, the authoritative approach is worth recommending all over the world [100]. Furthermore, while analysing the relationship of parental educational status with the academic achievements of the students, the authors reported that as compared to students whose parents

are qualified below 12 grade, those students performed 20 points higher than whom parents are qualified above grade 12. Similarly, the availability of assets at the home is also directly associated with the achievements of the students in a science subject [101].

The socioeconomic status is the combination of a variety of factors out of which monthly family income, literacy levels of parent and their occupation or sources of income are the most important indicators of academic achievements. The family socioeconomic status can facilitate the academic achievements of the students in three important ways i.e. provision of financial and material support, timely and appropriate guidance from paid sources and safer and elevated social status than that is possessed by the rest of the society [102, 103]. The authors assessed the South African learners' performance in mathematics and found that not only the availability of facilities at the school but also parental educational level is one of the strongest predictors of students' academic achievements. However, it was found that the majority of the students perform academically averagely irrespective of the socioeconomic standings of the family [104]. One of the studies from Nigeria highlighted that students from low socioeconomic status usually perform lower academically as compared to those from higher socioeconomic status families [105]. Similarly, another study in Nigeria reported that parental deprivation caused poor provision of learning materials and a poor learning environment to children resulting in their poor academic performance [106]. Children from low socioeconomic status families struggle hard in achieving their developmental goals. A child from low socioeconomic status, despite of their high capabilities, are more likely to secure low grades than a child of the same capabilities but from a higher socioeconomic background. Socioeconomic status is also influential in raising the mental and emotional well-being of children which positively affects their academic performance [107].

In most of the patriarchal societies in developing nations, gender is also an important predictor of family interest in spending on children's education. For instance, boys are preferred over girls in education in male-led societies. Due to this differential care of males in terms of positive attitude towards their education and high educational spending on them, the literacy levels and grades may vary on the basis of the gender of the students [108, 109]. Gender-based variation in the educational performance of children may vary according to parenting style, parental involvement and other micro and meso-level systems due to varying socio-cultural reasons [110, 111], some factor (variables) are more favorable to boys and other to girls [112, 113]. Experiments on the psychological ground, however, consistently show little significance of gender differences in the cognitive abilities of children. Therefore, in egalitarian societies, cognitive abilities are an important interpersonal character that is obviously linked to the academic achievement of children. However, in other societies, socioeconomic and cultural factors are more important in bringing variations in the academic performance of children based on their gender [114].

## Study methodology

### Study design

The research design for this study was a "Cross-Sectional" or one-shot or status design based on both times of exploration and study population. This study design is most appropriate for knowing the existing phenomenon, problem, attitude, perception, or issue, by taking a cross-section of the population [115].

### Sample size and sampling

The study was carried out in District Malakand, a rural district in Malakand Division of Khyber Pakhtunkhwa province (Pakistan). The District comprises of two Tehsils and 28 Union

 

Councils (UCs) with a total area of 952 square km. According to 2017 population census the total population of the district was 717806 (90.57% rural and 9.43% urban population) with a population density of 750/square km. Pushto is the main language of the 98.2% residents of the District. The District was formed as a Provincially Administered Tribal Area (PATA) in 1970, earlier to which it was a tribal area known as Malakand protected area since the British rule. For selecting a representative sample, purposive sampling method with specific criteria for selection was adopted in the following manner [116].

District Malakand and its two Tehsils were purposively selected at the first and second stages. The two tehsils, comprising 28 union councils (UCs), included five urban and 23 rural union councils with a ratio of 1:5. Using this logic, two urban and ten rural union councils were randomly selected at third stage. Out of each selected UC one government boy's school, one government girl's school and one private school were randomly selected to collect information at the fourth stage.

As per office record of the selected schools, the total number of students in the 9th and 10th classes in the selected 36 schools was 7,952 students, out of which 6,701 students were from government schools and 1251 from private schools, moreover, 3959 among these students were girls and 3,993 boys. Total required sample size for 7952 students was worked out as 448 (Eq-I) and proportionally allocated (Eq II) to each school (Table 1) according to the number of secondary level students (class 9 and 10) in them [117, 118].

$$n = \frac{N\hat{p}\hat{q}Z^2}{\hat{p}\hat{q}Z^2 + Ne^2 - e^2} \tag{Eq-I}$$

Where, N = total number of students in 9th and 10th class in selected secondary schools = 7952, p = population proportion = 0.50, q = opposite proportion q = (1-p) = 0.50, z = confidence level = 1.96, e = margin of error = 0.045, n = 448.

$$n_h = (N_h/N) * n \tag{Eq-II}$$

Where $n_h$ is the sample size for stratum $h$, $N_h$ is the population size for stratum $h$, and N is total population size, and n is the total sample size

To meet the study objectives students of 9th and 10th classes of the age group 15–18 years, including boys and girls, were randomly selected from government and private schools as study respondents.

The current study comprises of three independent variables namely (1) parenting style (authoritative, authoritarian and permissive), (2) student's gender and (3) family's socioeconomic status (low and high socioeconomic status). The dependent variable was the children's academic performance (Table 2).

**Table 1. School type, location, students' enrolment and required sample size.**

| School Type | Location a wise number of selected schools | | Enrolment | | Total number of students | Sample Size |
|---|---|---|---|---|---|---|
| | Urban | Rural | 9th class | 10th class | | |
| Government Boys | 2 | 10 | 1805 | 1435 | 3240 | 183 |
| Government Girls | 2 | 10 | 1799 | 1662 | 3461 | 195 |
| Private | 2 | 10 | 693 | 558 | 1251 | 70 |
| Sub Total | 6 | 30 | 4297 | 3655 | 7952 | 448 |

**Table 2. Conceptual framework.**

| Independent Variables | Dependent Variable |
|---|---|
| | Children academic Performance |
| Parenting styles (authoritative, authoritarian, permissive) Family socioeconomic status Gender | |
| | |

## Measurement of variables

Measurement of parenting style was based on the methodology developed by Robinson et al. (1995). The parenting style scale consisted of three subscales i.e. authoritative parenting (eight items), authoritarian parenting (six items) and permissive parenting (four items). All three parenting style variables scales were internally consistent and exhibited Cronbach's alpha coefficient value of 0.76, 0.72 and 0.78 respectively [119], therefore, the variables were indexed and correlated with each other to ascertain correlation among the parenting styles subscales. As all three subscales of parenting styles were found negatively correlated so for the current study all three subscales were considered as a single parenting style scale (Table 3). Each respondent was ranked into a specific parenting style category based on their highest average score [120].

The measurement scale for family socioeconomic status (SES) of the students was designed by following Kuppuswamy's modified socioeconomic (SES) scale which is based on the composite score of the three important variables that are parental education, family monthly income and occupation/income source. The SES scale was internally consistent and suitable for indexation with Cronbach's alpha value of 0.79. The domain of parent education was scored on seven levels (1 = illiterate, 2 = primary (five years of education), 3 = middle (eight years of education), 4 = secondary/matriculate (ten years of education), 5 = intermediate (twelve years of education), 6 = bachelor degree (fourteen years of education), and 7 = masters and above (sixteen years of education and above)). The domain of family monthly income was scored on three levels (1 = monthly income Up to PKRs. 250000, 2 = monthly income PKRs. 260000 to 50000and 3 = monthly income PKRs. 510000 and above). The domain of occupation/major income source was measured on four levels scale (1 = private job/business, 2 = agriculture, 3 = remittances and 4 = government job). Based on the scores level, the highest possible score of the three mentioned domains measuring socioeconomic status (SES) of the family became 14. Thus respondents with a score of 7 or below on the socioeconomic status scale were ranked into Low Socioeconomic status families and those scoring above 7 on the scale were considered as from High Socioeconomic status families [121].

Students' academic performance was based on percent marks in the last exam. For hierarchical multiple regression analysis, the academic score of respondents in percentages was considered. Whereas, for the application of non-parametric tests (chi-square and Kendall's Tau-c tests) the variable of the academic score was divided into four categories: (1) A1 grade (80% and above marks), (2) 1st division (60% to below 80% marks), (3) 2nd division (45% to below

**Table 3. Association between parenting style and children academic performance.**

| Parenting style | Academic performance | | |
|---|---|---|---|
| | Chi-square Value | P-Value | Tau-c |
| Authoritative parenting | 273.605 | 0.000 | 0.500 |
| Authoritarian parenting | 166.858 | 0.000 | -0.369 |
| Permissive parenting | 235.651 | 0.000 | -0.030 |

60% marks) and (4) 3rd division and below (below 45% marks). These categories cross-tabulated with parenting style to test the association of academic performance of children with parenting style.

### Dummy coding of the variables

Dummy coding of variables for multivariate analysis is given as follows. Moreover, the actual percentage score of the students during the last exam was used in Hierarchical multiple regression as a measure of the academic performance of students (dependent variable).

### Dummy coding of the variables

The categorical variables of the study were dummy coded in the following fashion.

| | |
|---|---|
| Socioeconomic status (SES) | 0 if low SES, 1 otherwise |
| Parenting Style | 0 if authoritarian, 1 if permissive, 2 if authoritative |

## Data analysis

The collected data was coded and entered in SPSS software for its analysis. The chi-square test and Kendall's Tau-c tests were applied to test the strength and direction of the association of parenting style with the academic performance of children while keeping student gender and family socioeconomic status as a control variable. Moreover, Fisher's Exact test was introduced as an alternative to the Chi-square test where the condition of the Chi-square test (sufficiently large sample size that no expected frequency is less than five) was violated [122].

### Hierarchical multiple regression

Theoretical foundations for conducting hierarchal multiple regression analysis are provided by Chiu & Chow and Masud and his colleagues who envisaged that prediction of academic performance of children based on parenting style may vary significantly with the socioeconomic status of child's family. These theoretical grounds were used in hierarchical multiple regression analysis to determine how much the parenting style and its interaction with family socioeconomic status account for the academic performance of the children [123, 124].

### Assumptions testing for hierarchical multiple regression

Using the statistical procedures, the basic assumptions such as sufficiency of required sample size, normality, linearity, multicollinearity of variables and homoscedasticity were tested and confirmed for the robustness of the models for running hierarchal multiple regression.

### Variables selection

A stepwise method was used for the selection and ordering of variables in the hierarchical regression model [125, 126]. It helped to determine the most important predictors of the academic performance of children and the ordering of the variables for entry in the hierarchical regression model based on their standardized Beta ($\beta$) coefficients without significantly reducing the R-square ($R^2$) coefficient of the model. The stepwise variable selection technique also helped to validate the theory put forward by Beauvais and his colleagues and Tomul regarding the strength and level of the importance of the child's background (socioeconomic status) and micro level predictor (parenting style) in relation to the academic achievements of a child [127, 128].

Statistical significance of the Hierarchical regression model at each stage (stages 1–5) and of the full model (stage 5) was tested by using F-test. Statistical significance of independent variables in predicting dependent variables was tested by using a t-test.

### Ethical approval statement

This research study followed the APA standards of ethical considerations for conducting research which emphasizes preserving the dignity and humanity of respondents, avoiding harm, anxiety, distress, pain, or any negative feeling of the research subjects. The study proposal was approved by the Advanced Studies & Research Board of the university. And before going to data collection the interview schedule was pretested to double-check the instrument for ethical issues and the problems were removed. Also, the research was conducted under the authority letter from the university clearly explaining that the information collected will be used for research and academic purpose only. Similarly, an authority letter was signed by District education departments (both male and female) before starting data collection from secondary school students. Written informed consent was obtained from schools head by providing the signed authority letter from district education departments (Male and Female) prior to data collection.

An interview protocol was devised to assist getting answers on questions asked from the respondents. Besides questions on main study variables, the interview protocol included the script of introductory para before interview, explanatory para for each study variable, and a script for concluding interview. For this purpose, after formal permission from school heads, an informed verbal consent was obtained from secondary school students in presence of their teachers. The students who were willing were interviewed in the presence of their teachers and in the interview schedule only questions were included related to the study variables and no such questions were included that were related to the students identity and they were satisfied of their confidentiality before providing their responses. The respondents were free to terminate their interview at any stage. The students were briefed about parenting styles beforehand asking questions on the variables. In addition, keeping into consideration the cultural reasons, the data from female respondents were collected through a trained female investigator under the guidance of the researcher.

### Limitations of the study

The study was limited to only two levels, Micro- and Meso-levels, of Bronfenbrenner's socioecological model and focused on the influence of students' gender, parenting style and family's socioeconomic status on the academic performance of secondary school students.

## Results

### Association between parenting style and children academic performance

Results in Table 3 unveil that the association of authoritative parenting style with children's academic performance was highly significant (P = 0.000) and positive ($T^c$ = 0.500). While the association of authoritarian parenting style with children's academic performance was highly significant (P = 0.000) and negative ($T^c$ = -0.0.369). Similarly, the association between permissive parenting style and children's academic performance was found highly significant and negative (P = 0.000, $T^c$ = -0.030).

### Association between parenting style and children's academic performance (after controlling for the family socioeconomic status of the respondents)

Results in Table 4 show that the influence of parenting style on the academic performance of children in the context of respondents' socioeconomic status was positive ($T^c$ = 0.689) and

**Table 4.** Association between parenting style and children academic performance (controlling family socioeconomic status of the respondents).

| Family socioeconomic status (SES) | Independent Variable | Dependent Variable | Statistics X2, (P-Value) & $T^c$ | Statistics X2, (P-Value) & $T^c$ for entire table |
|---|---|---|---|---|
| High SES | Parenting style | academic performance of children | χ2 = 27.433 (0.000) $T^c$ = 0.689 | χ2 = 42.303 (0.000) $T^c$ = 0.487 |
| Low SES | Parenting style | academic performance of children | χ2 = 20.829 (0.009) $T^c$ = 0.218 | |

highly significant (P = 0.000) among the above-mentioned variables for respondents from high socioeconomic status. The association of the above-said variables was also positive ($T^c$ = 0.218) and significant (P = 0.009) for respondents from low socioeconomic status. The value of significance and $T^c$ for the entire table show a highly significant and positive association (P = 0.000 & $T^c$ = 0.487) between parenting style and academic performance for respondents from both high and low socioeconomic statuses.

## Association between parenting style and children's academic performance (after controlling for the gender of the respondents)

Results in Table 5 showed that the influence of parenting style on the academic performance of children in the context of respondents' gender was positive ($T^c$ = 0.129) and significant association (P = 0.002) in the above-mentioned variables for males. The association of the above-said variables was also positive ($T^c$ = 0.139) and significant (P = 0.008) for female respondents. The value of significance and $T^c$ for the entire table show a highly significant and positive association (P = 0.003 & $T^c$ = 0.123) between parenting style and academic performance for both genders. The similar $T^c$ chi-square values indicate that the effects of parenting style on the academic performance of children were almost identical for both genders. Therefore, parenting style emerged as a universal factor in influencing the academic performance of children irrespective of the gender of the respondents.

## Hierarchical multiple regression analysis

Preceding the statistical procedure for hierarchical multiple regression, the sufficiency of the required sample size was determined. In addition, four basic assumptions (normality, linearity, multicollinearity and homoscedasticity) of the linear regression model were tested.

## Assumptions testing for hierarchical regression analysis

**Sufficiency of required sample size.** To determine whether the sample size of 448 is sufficient for five variables that were subjected to test their ratios that came out as 1:64 which fulfilled the minimum required ratio of 1:20. Thus, the condition of the sufficiency of sample size was satisfied [129]. Therefore, the sample size was sufficient for running Hierarchical multiple regression.

**Table 5.** Association between parenting style and children academic performance (controlling gender of the respondents).

| Gender | Independent Variable | Dependent Variable | Statistics χ2, (P-Value) & $T^c$ | Statistics χ2, (P-Value) & $T^c$ for entire table |
|---|---|---|---|---|
| Male | Parenting style | Academic performance of children | χ2 = 41.918 (0.002) $T^c$ = 0.129 | χ2 = 68.253 (0.003) $T^c$ = 0.123 |
| Female | Parenting style | academic performance of children | χ2 = 35.484 (0.008) $T^c$ = 0.139 | |

**Table 6. Normality diagnostics test.**

| Kolmogorov-Smirnov[a] | | | Shapiro-Wilk | | | Skewness (Standard error) | Kurtosis (Standard error) |
|---|---|---|---|---|---|---|---|
| Statistic | df | Sig. | Statistic | Df | Sig. | | |
| 0.023 | 448 | 0.200* | 0.998 | 448 | 0.883 | 0.059 (0.115) | -0.041 (0.230) |

**Normality diagnostics.** Results of the normality test in Table 6 highlight that the values for skewness (0.059) and kurtosis (-0.041) satisfy the minimum requirements of +/-0.5 [130]. Furthermore, the assumption of normality was satisfied at a P = 0.05 level of significance, as the significance values for both of the Kolmogorov-Smirnov and Shapiro-Wilk tests were (0.2000 and 0.883) respectively (P>0.05) as presented in Table 6. Thus, the hypothesis of abnormal distribution of data was rejected and the normal distribution of academic scores of students was confirmed.

**Linearity.** The P-P Plot (Fig 1) represents the relationship between the independent variables and dependent variable (academic performance). The P-P Plot looks pretty linear with only a few points positioned below or above the (imaginary) straight line standing out a perfect linear relationship, so the condition of linearity was also satisfied [131].

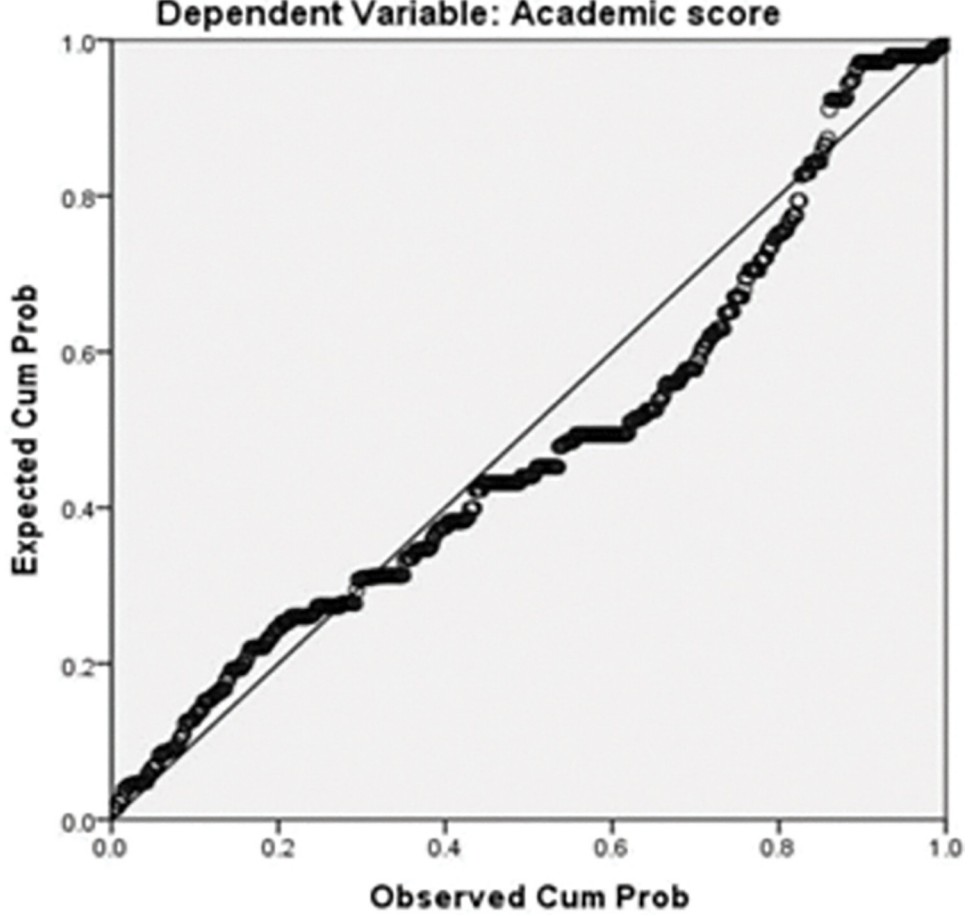

**Fig 1. Normal P-P plot of regression standardized residuals.**

**Table 7. Correlation between variables.**

| S. No | Variable | 1 | 2 | 3 | 4 | 5 |
|---|---|---|---|---|---|---|
| 1 | Socioeconomic status (SES) | ____ | | | | |
| 2 | socioeconomic status & authoritative parenting | 0.673** | ___ | | | |
| 3 | Authoritative parenting | 0.513** | 0.672** | ___ | | |
| 4 | socioeconomic status & permissive parenting | 0.113* | -0.226** | -0.265** | 0.058$^{NS}$ | |
| 5 | Permissive parenting | -0.139* | -0.325** | -0.381** | -0.008$^{NS}$ | 0.695** |

*Note*. Data in table give correlation coefficient (α) value while * represent significant (P≥0.01 and ≤ 0.05), ** represent high significant (P = 0.000) and NS represent non-significant correlation (P>0.05)

**Multicollinearity diagnostics.** The bivariate correlation Table 7 shows that the correlation between all the predictors' variables (1) family socioeconomic status (SES) (2) interaction of socioeconomic status & authoritative parenting (3) authoritative parenting (4) interaction of socioeconomic status and permissive parenting, and (5) permissive parenting were statistically significant as ($P < 0.05$) and their values range between $r = -0.381$ and $r = 0.695$ which were below the permissible limit of $r = 0.7$. Moreover, the VIF values for all of the predictor variables are smaller than 10, and also, all tolerance values were higher than 0.10 (Table 10). These results illustrate that there was a lesser chance of multicollinearity between the predictor variables so the condition of multicollinearity was satisfied [132, 133].

**Homoscedasticity.** The scatterplot (Fig 2) represents the dependent variable (academic score) with regression standards residuals. The scatterplot shows that none of the points of the Scatterplot falls outside of -3 and to 3 on X-axis or Y-axis. Therefore, the condition homoscedasticity was also satisfied and the data is suitable for Hierarchical multiple regression [134].

Since all four assumptions (multicollinearity, normality, linearity, and homoscedasticity) along with the sufficiency of the required sample size have been satisfied, the dependent and independent variables are adequate for performing hierarchical multiple regression analysis.

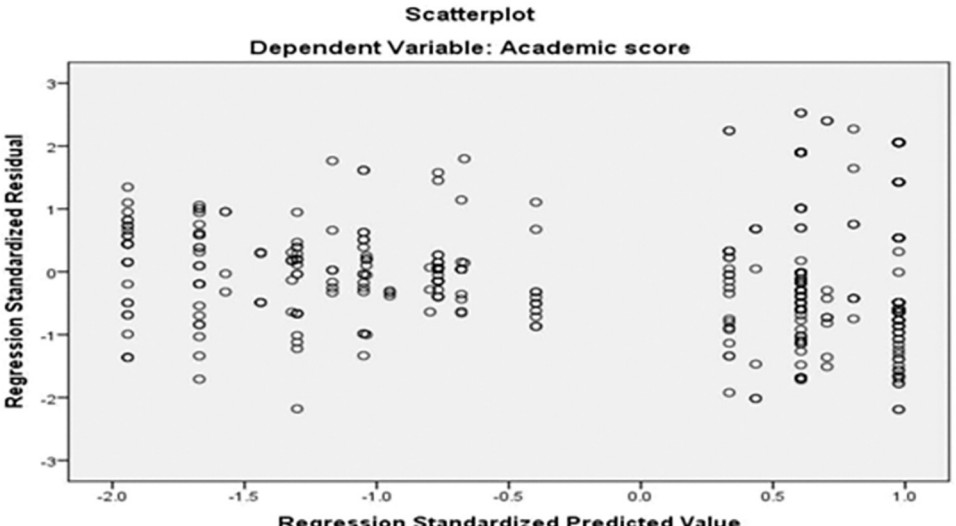

**Fig 2. Scatter plot of academic score with regression standardized residuals.**

### Hierarchical multiple regression analysis results

#### Variable selection

All of the five predictor variables (family socioeconomic status, authoritative parenting, permissive parenting, the interaction of socioeconomic status and authoritative parenting, and interaction of socioeconomic status and permissive parenting) proved significant ($P < 0.05$) in the Hierarchical multiple regression model to explain variation in the dependent variable. Hence, all of the five variables were introduced in the hierarchical regression model.

#### Analysis of variance (ANOVA)

Analysis of variance (ANOVA) was conducted to test the statistical significance of the overall model. The Analysis of variance (ANOVA) results in (Table 8) show that the models as a whole (which includes all the five models or blocks of variables) is statistically significant ($P < 0.05$, F (df 7 = 5, 442 = 128.848)) which illustrate that the independent variables significantly predict the dependent variable (Academic performance of children). Hence, the hierarchical regression model was a good fit for the data.

#### Model summary results

A five-stage Hierarchical Multiple Regression was conducted to test the relationship between the set of independent variables (family socioeconomic status, authoritative parenting, permissive parenting, the interaction of socioeconomic status and authoritative parenting, and the interaction of socioeconomic status and permissive parenting) and the dependent variable (academic performance of children) (Table 9).

The family socioeconomic status (SES) was entered at stage one, the interaction effect of socioeconomic status & authoritative parenting in the second stage, authoritative parenting at the third stage, the interaction of socioeconomic status & permissive parenting at the fourth

**Table 8. ANOVA.**

| Model | | SS | Df | MS | F | Sig. |
|---|---|---|---|---|---|---|
| 1 | Regression | 121864.711 | 1 | 121864.711 | 386.777 | 0.000[a] |
| | Residual | 140524.689 | 446 | 315.078 | | |
| | Total | 262389.399 | 447 | | | |
| 2 | Regression | 143752.314 | 2 | 71876.157 | 269.603 | 0.000[b] |
| | Residual | 118637.086 | 445 | 266.600 | | |
| | Total | 262389.399 | 447 | | | |
| 3 | Regression | 147095.501 | 3 | 49031.834 | 188.823 | 0.000[c] |
| | Residual | 115293.898 | 444 | 259.671 | | |
| | Total | 262389.399 | 447 | | | |
| 4 | Regression | 154549.962 | 4 | 38637.490 | 158.721 | 0.000[d] |
| | Residual | 107839.437 | 443 | 243.430 | | |
| | Total | 262389.399 | 447 | | | |
| 5 | Regression | 155621.046 | 5 | 31124.209 | 128.848 | 0.000[e] |
| | Residual | 106768.354 | 442 | 241.557 | | |
| | Total | 262389.399 | 447 | | | |

*Note.* [a]Denote relationship of Socioeconomic status (SES) and academic performance. [b]Denote relationship of the interaction between socioeconomic status & authoritative parenting with and academic performance. [c]Denote relationship of Authoritative parenting and academic performance. [d]Denote relationship of the interaction socioeconomic status & permissive parenting and academic performance. [e]Denote relationship of Permissive parenting and academic performance.

**Table 9. Model summary.**

| Model, Step, and Predictor Variables | $R^2$ | SE | $(\Delta R^2)$ | $(\Delta F)$ | df | Sig |
|---|---|---|---|---|---|---|
| Model 1<br>1-Socioeconomic status (SES) | 0.464 | 17.75043 | 0.464 | 386.777 | (1, 446) | 0.000 |
| Model 2<br>1-Socioeconomic status (SES)<br>2-SES & authoritative parenting | 0.548 | 16.32790 | 0.083 | 82.099 | (1, 445) | 0.000 |
| Model 3<br>1-Socioeconomic status (SES)<br>2-SES & authoritative parenting<br>3-Authoritative parenting | 0.561 | 16.11431 | 0.013 | 12.875 | (1, 444) | 0.000 |
| Model 4<br>1-Socioeconomic status (SES)<br>2-SES & authoritative parenting<br>3-Authoritative parenting<br>4-Neighborhood services | 0.589 | 15.60224 | 0.028 | 30.623 | (1, 443) | 0.000 |
| Model 5<br>1-Socioeconomic status (SES)<br>2-SES & authoritative parenting<br>3-Authoritative parenting<br>4-Neighborhood services<br>5-SES & permissive parenting | 0.593 | 15.54212 | 0.004 | 4.434 | (1, 442) | 0.036 |

stage and permissive parenting at the fifth stage. Stepwise and sequential inclusion of the above variables in hierarchical regression models was based on theoretical grounds proposed by Pant and Mwariri and colleagues [135, 136]. Furthermore, the weighted importance of each variable in influencing the academic performance of children, as determined through the standardized Beta coefficient (β) values (Table 10) also validated the above sequence for entry of variables in hierarchical regression models.

The hierarchical multiple regression model summary (Table 9) reveals that at stage one (model-1), family socioeconomic status (SES) significantly contributed to the regression model (P<0.05) and accounted for 46.4% (R2 = 0.464) of the variance in children's academic performance.

Introducing the interaction of socioeconomic status & authoritative parenting at the second stage (model-2) explained 54.8% variance in the academic performance of children ($R^2$ = 0.548) and significantly contributed to the regression model (P<0.05). The $R^2$ change in model-2 indicates that the additional 8.4 percent variance in the academic performance of children is explained by the interaction of socioeconomic status & authoritative parenting as compared to that was explained (46.4%) by only family socioeconomic status in model-1.

**Table 10. Coefficient.**

| | Unstandardized Coefficients | | Standardized Coefficients | t | Sig. | 95% confidence interval for B | | Correlations | | | Collinearity statistics | |
|---|---|---|---|---|---|---|---|---|---|---|---|---|
| | B | SE | Beta | | | Lower Bound | Upper Bound | Zero-order | Partial | Part | Tolerance | VIF |
| (Constant) | 36.317 | 1.819 | | 19.965 | 0.000 | 32.742 | 39.892 | | | | | |
| Socioeconomic Status (SES) | 18.252 | 2.946 | 0.335 | 6.196 | 0.000 | 12.463 | 24.042 | 0.681 | 0.283 | 0.188 | 0.315 | 3.173 |
| SES*authoritative parenting | 14.188 | 4.111 | 0.288 | 3.451 | 0.001 | 6.108 | 22.268 | 0.698 | 0.162 | 0.105 | 0.132 | 7.548 |
| Authoritative parenting | 13.384 | 3.260 | 0.260 | 4.105 | 0.000 | 6.977 | 19.791 | 0.608 | 0.192 | 0.125 | 0.229 | 4.361 |
| SES*permissive parenting | 11.464 | 6.394 | 0.122 | 2.575 | 0.010 | 3.897 | 29.030 | 0.093 | 0.122 | 0.078 | 0.408 | 2.454 |
| Permissive parenting | 9.278 | 4.406 | 0.096 | 2.106 | 0.036 | .619 | 17.937 | -0.058 | 0.100 | 0.064 | 0.445 | 2.250 |

*Note.* *Denote interaction between independent & dependent variables, unstandardized B coefficient denote strength of the influence of independent in the dependent variable, P<0.005 denote significant association

The addition of authoritative parenting to the regression model at the third stage (model-3), significantly contributed (P<0.05) and accounted for 56.1 percent variance in children's academic performance (R2 = 0.561). Moreover, the $R^2$ change highlighted that the additional 1.3% variance is explained by authoritative parenting in the academic performance of children which was accounted for by the variables in model-2 (family socioeconomic status and interaction of socioeconomic status & authoritative parenting).

Adding interaction of socioeconomic status and permissive parenting at the fourth stage (model-4) significantly contributed (P<0.05) and accounted for a 58.9% variance in children's academic performance (R2 = 0.589). The $R^2$ change showed that the additional 2.8% variance is explained by the introduction of the interaction of socioeconomic status and permissive parenting to the variables entered in model-3 (family socioeconomic status, interaction of socioeconomic status & authoritative parenting, and authoritative parenting).

Finally, the addition of permissive parenting at the fifth stage (model-5), significantly contributed (P<0.05) and explained 59.3% variance ($R^2$ = 0.593) in children's academic performance. $R^2$ change depicts that the additional 0.4% variance is explained by the introduction of permissive parenting to the variables entered in model-4 (family socioeconomic status, interaction of socioeconomic status & authoritative parenting, authoritative parenting and interaction of socioeconomic status & permissive parenting).

All of the above five regression models are statistically significant (P < 0.05), meaning that the introduction of an additional variable(s) at each step/block produces a statistically significant increase in variance accounting for the dependent variable i.e. (academic performance of children).

## Coefficient

Results in Table 10 show that all the five predictor variables introduced in the models (socioeconomic status (SES), authoritative parenting, permissive parenting, interaction between socioeconomic status and authoritative parenting and interaction between socioeconomic status and permissive parenting) have statistically significant (P<0.05) impact on academic performance of children.

Furthermore, values of unstandardized coefficients (Table 10) indicate that by keeping all variables constant, a student is expected to secure 36.31% (β = 36.31) marks in exams as indicated by the constant value of the coefficient. Moreover, children from higher socioeconomic status (SES) families secure 18.25% higher marks (β = 18.25) compared to children from low socioeconomic status (SES) families. In addition, the interaction of socioeconomic status and authoritative parenting have a significant effect on the academic performance of children for a unit change (low to high) in socioeconomic status 14.18% increase occurs in the academic score of children under the authoritative parenting style compared to authoritarian parenting (β = 14.18). Moreover, authoritative parenting independently increases the academic score of children by 13.38% relative to authoritarian parenting (β = 13.38). Likewise, the interaction of socioeconomic status and permissive parenting have a significant effect on the academic performance of children as a unit change (low to high) in socioeconomic status 11.46% increase occurs in marks of children under permissive parenting compared to authoritarian parenting (β = 11.46). Similarly, permissive parenting independently increased the academic score of children by 9.27% relative to authoritarian parenting (β = 9.27).

The hierarchical regression model specification for this study is given as follows: Y (Academic performance of children) = 36.31 + 18.25 (socioeconomic status) + 14.18 (socioeconomic status *authoritative parenting) + 13.38 (authoritative parenting) + 11.46 (socioeconomic status *permissive parenting) + 9.27 (permissive parenting).

## Discussion

The current study was conducted for three basic purposes. The first purpose of the present study was to investigate the association of parenting style with children's academic performance. Secondly, to examine the association of parenting style with children's academic performance on the basis of their gender and family socioeconomic status and thirdly, to assess the level of the effects of the group variables (parenting style and family socioeconomic status) on children's academic performance independently and in interaction with each other.

The association of parenting style with children's academic performance (Table 3) indicates that the authoritative parenting style is significantly and positively associated with the academic performance of children compared to permissive and authoritarian parenting. Authoritative parenting is characterized by high responsiveness and high demandingness. These parents are more responsive towards their children give high value to warm and timely guidance to their children when needed. The authoritative parents are warmer and less demanding, and show more responsiveness towards children which leads to more confidence and higher engagement in their academic tasks ultimately resulting in higher academic achievements. Children of such parents are found regular in their school attendance, do timely homework and are found responsible for preparing themselves for their exams. Thus, the democratic culture promotes an authoritative parenting habits among parents, that groom children to become responsive to their work and ensures their better academic performance in a positively competitive educational environment. Similar findings are concluded by Grant & Ray and Steinberg that authoritative parents keep a keen eye on their children's deviant behaviour and bring them to conformity while emphasizing responsiveness and seldom punitive remedies. Such a positive parent-child relationship is based on trust, understanding and caring which enhance the motivational level of the child with better academic achievements [137, 138]. Checa maintained that authoritative parents keep an adequate balance of autonomy and responsibility for their children according to the situational requirements [139]. Children from such families are provided with opportunities to share their views on family issues through participation in family decisions, discuss their concerns and develop a positive relationship with parents and other family elders. The children also develop a sense of being valuable member of the family and that their views are important for family decisions. In addition, the children also know the rules and values established at their family that delineate the limits of their behaviour which may result in imposing sanctions on them in case of crossing the limits. The author added that authoritative parenting is stimulating to the child's self-efficacy, which has a significant contribution to their academic performance. Thus, the extent to which authoritative parents encouraged children's autonomy and responsibility were important predictors of academic achievements. The strength of such prediction further increases when the authoritative parents were generous in spending on children's needs and regularly contacted teachers to get feedback from them regarding children's education and their overall development [140].

The authoritarian parenting style is more focused on disciplining children with less heed to the child's participation and social needs. The parents adopt psychological and physical punishments to discipline their children ranging from mild punishments (taking away privileges from children and reminding parental efforts for child welfare) to moderate and even extremely harsh punishments (yelling, threatening, open criticism and spanking). Mild punishments have a weak negative influence on the academic performance of children. However, harsh and violent physical and psychological punishments exhibit strong negative impacts and are thus linked with poor academic performance of children. The gravity of harsh punishments and their regular occurrence overburden the child psychologically resulting in their

lowered learning abilities, loss of interest in educational and other normal life activities and overall poor academic performance in them. The authoritarian parenting style is linked to high demandingness and low responsiveness because such parents control the behaviour of their children through an absolute set of principles with little value for responsiveness, and warm and timely guidance. These parents lay high emphasis on bringing conformity in their children by using a variety of extreme disciplinary measures ranging from simple disapproval to psychological and physical punishment and torture. Thus, fear of punishment is a more important reason to direct children towards a particular goal than other positive motivational reasons. Authoritarian parents try to control the behaviour of their children with greater emphasis on punitive strategies including physical, psychological and socioeconomic punishments. The authoritarian parenting style is characteristic of patriarchal societies where parenting is strict and deviation from rules and regulations is unbearable. Any mistake or failure of the children is treated with disciplinary measures with little provision of guidance or counseling from parents [141, 142]. The negative and positive influences of authoritarian parenting vary across cultures. In societies with strong cultural holds of patriarchy and with more emphasis on subordination and obedience, authoritarian parenting is correlated with some moderate level of promising educational outcomes in children. However, despite their better academic grades, children from authoritarian families exhibit low communication skills and low capabilities of self-expression than those children from authoritative families [143, 144].

The permissive parents ignored the minor mischiefs and misbehaviour of their children, however, the negative effects of ignoring petty mischiefs of children on their academic performance was non-significant. In some extreme cases, the parents lose their control over children which makes it difficult for them to discipline their children. Such children, at some point, start to realize that their parents, due to their lenient behaviour, have spoiled them. This aggravated situation adversely affects the academic performance of children in connection with permissive parenting. In the permissive parenting style presented by Baurmind, the parents exhibit a warm response to their children in a way that they permit their children whatever they want to do with little or no control over them. The above-mentioned three parenting style typology devised by Burmind and explained by Maccoby and Martin is adopted for this study. In permissive parenting, the parents have at least one common characteristic that they are low demanding from their children and literally overlook any disciplinary measure required to bring their children to conformity. However, within permissive parenting, the parents may vary on the basis of the degree of responsiveness (high to low responsiveness). Thus, children from permissive families lack guidance from their parents. Most of these children are spoiled due to limitless freedom and insufficient guidance [145]. Empirical studies have negatively linked the academic performance of children to permissive parenting with some variations according to the culture and interpersonal traits of the child [146].

Similarly, Cherry carried out a research study in South Africa which indicated a significant association between parenting style and the academic performance of students [147]. Also, parenting style as one of the correlates of academic performance [148]. Moreover, Tilahun revealed a negative association between permissive parenting and students' academic performance in Nigeria [149].

Furthermore, the association between parenting style and children's academic performance is spurious on the basis of respondents' socioeconomic status (Table 4). Respondents from high socioeconomic statuses, when exposed to a similar parenting style, secured better school grades than children from low socioeconomic groups. The probable reason for these findings is that the complementary effect of socioeconomic status in terms of meeting the basic life and educational needs of the children. In this way, children from high socioeconomic status and appropriate parenting are in a better position to secure better grades than children with similar

parenting but a low socioeconomic group. Therefore, a combination of favourable parenting with high socioeconomic status is a better predictor of academic performance than better parenting alone. Ashiona and Teresa in their research in Kenya reported the association between family socioeconomic status and admission in schools and colleges. The authors reported that children from well-off families secured admission in high-quality educational institutions and ultimately performed better in exams [150]. Parental education, occupation and income are important determinants of socioeconomic status. Due to better education and high income, parents in high socioeconomic groups are in a good position to protect their children from risks [151]. Parents' good educational background and high socioeconomic status are fruitful for the educational guidance of their children. Moreover, their sound income standing helps children in availing better educational facilities [152]. Conversely, positive parenting may not be that beneficial when financial support becomes a limiting factor for children's education. Similarly, the illiteracy of a caring parent become a constraint to correctly understand the educational need of a child. Thus, the socioeconomic status of parents affects the parenting style that subsequently affects the education of the children [153].

Gender is an important divide that influences life achievements among male and female children. Children, based on their gender, receive varying treatment in patriarchal societies. This also stands true for cross-gender attachment between parents and children where the mother offers great care to their sons while, the father gives due attention to their daughters. Therefore, discriminatory exposure of children based on their genders to different parenting styles may lead to their varying academic performance [154]. However, exposure of children to a similar parenting style without gender-based discrimination is likely to result in almost similar academic performance of the children irrespective of their gender [155]. A shift in mindset of parents towards a more egalitarian treatment to children, especially with respect to their education, is the important reason to explain a balance in educational performance of children from both genders, irrespective of their parenting style.

A series of studies linked the academic performance of children with family socioeconomic status (SES) and parenting style [156, 157]. In the hierarchical multiple regression analyses in the family, socioeconomic status emerged as the most important factor in the academic performance of children as reported by several researchers [158, 159]. This important variable touches on each and every aspect of education and learning to bring substantial variation in the academic performance of children. The high socioeconomic status of the family ensures quality child care, safety and spending on their basic and educational needs which have a direct relationship with the academic performance of children. The provision of these facilities can help the children to catch up with the high capability of children from low socioeconomic background [160]. The positive effects of high socioeconomic status are so vital that it masks the negative influences of inappropriate parenting and the same is evident from the interaction effect of socioeconomic status with permissive parenting in the current study. The socioeconomic status also accounts for cultural differences in parenting style. Thus, some parents with permissive or authoritarian styles manage to adopt positive parenting traits due to high socioeconomic status that promote better academic performance in children [161]. Therefore, high socioeconomic status and positive parenting is the best combination to ensure better academic grades in children [162–164].

This research adds that the strength of variables (parenting styles) to predict the academic performance of children is enhanced when they are applied in interaction with the socioeconomic status of the families. Thus, an improved socioeconomic status (high income, high parental education and highly prestigious occupation) in combination with a highly responsive and positively demanding parenting style (authoritative parenting) make the combination that promotes better academic performance in children. The academic performance of children is

faced with the highest setback due to the interactive effect of low socioeconomic status and inappropriate parenting (authoritarian and permissive parenting).

## Conclusion

It is concluded that children whose parents adopt authoritative parenting practices perform better academically. These parental attributes included responsiveness towards their children in terms of providing them with basic educational needs such as books, school uniforms, stationery items, taking care of the child's wishes while assigning them any task, encouraging the child to talk about their problems and giving due consideration to child preferences while making plans for the family. However, children experiencing authoritarian parenting (punishing children by taking privileges away from them, yelling, threatening, criticizing, spanking and reminding them of things parents do for children) and permissive parenting practices (ignoring children's deviant behaviour, rarely guiding or disciplining children, and spoiling children) exhibit low academic performance.

Moreover, the socioeconomic status of the child's family explains variation in his academic performance within a specific parenting style. Children from high socioeconomic status families are more likely to score high grades as compared to children from low socioeconomic families while experiencing the same parenting style. While there is no variation found in the academic performance of students based on their gender.

In addition, the study variables explained the academic performance of children in following order. Family socioeconomic status alone was the strongest predictor of academic performance of children, interaction of socioeconomic status and authoritative parenting was the second important predictor, authoritative parenting alone was third in importance, the interaction of socioeconomic status and permissive parenting stood at fourth place in importance, and permissive parenting ranked fifth in influencing academic performance of children in the study area.

Based on these conclusions, parenting style and socioeconomic status have important educational and economic policy implications regarding the formation of a positive attitude towards education among secondary school children and improving their academic performance, as highlighted in recommendations.

## Recommendations

Based on the findings of the study, the following recommendations are put forward:

The educational policy should include appropriate initiatives to enable and motivate parents through awareness and training programs for adopting appropriate parenting styles, with a special focus on parents from low socioeconomic status groups, to take care of the learning, educational and recreational needs of their children at home and provide them with the psychological and physical protection. The conducive learning environment so created will ensure better academic achievements in children.

The is a need to update federal policy for poverty eradication for its better implementation in letter and spirit to support low socioeconomic groups (having low income, unemployed and illiterate) for the quality school education of their children. Low socioeconomic standing of child's family was important determinants of unsatisfactory academic performance of children. These findings call for mobilization of the charitable resources like "Zakat" (religious binding on rich Muslims to distribute 2.5% of their financial resources like gold, silver and /or money among needy poor on annual basis) and "Ushr" (religious binding on Muslim farmers to distribute 1/10th to 1/20th of their agriculture produce among needy poor on each crop) for

supporting educational and other financial needs of families with low socioeconomic status. These social security measures are easy to enact due to strong religious endorsements.

Similarly, the government should provide scholarships to poor deserving children covering their educational and living expenses to overcome economic inequality as a cause of low school enrolment and performance. Such children would be able to spend maximum energy on their education and avoid child labour. Future studies are recommended on the effects of school type and the other two levels (Macro and Exo levels) of Bronfenbrenner's Socioecological model on academic performance.

## Author Contributions

**Conceptualization:** Nayab Ali, Asad Ullah, Aisha Khan, Rahat Ullah.

**Data curation:** Yunas Khan, Sajid Ali, Asad Khan.

**Formal analysis:** Nayab Ali, Asad Ullah.

**Investigation:** Mushtaq Ahmad.

**Methodology:** Nayab Ali, Asad Ullah, Abdul Majid Khan,  Bakhtawar.

**Project administration:** Umar Niaz Khan.

**Supervision:** Asad Ullah.

**Writing – original draft:** Nayab Ali, Abdul Majid Khan.

**Writing – review & editing:** Nayab Ali, Asad Ullah, Maaz Ud Din, Tariq Aziz.

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
