## [Decision Letter · Decision Letter 0]

14 Nov 2022

PONE-D-22-24697Academic performance of secondary school students in relation to parenting styles and family socioeconomic status: What attributes are important and how much?PLOS ONE

Dear Dr. Ali,

Thank you for submitting your manuscript to PLOS ONE. After careful consideration, we feel that it has merit but does not fully meet PLOS ONE’s publication criteria as it currently stands. Therefore, we invite you to submit a revised version of the manuscript that addresses the points raised during the review process. In addition, particular attention must be paid to the following comments. The article needs a thorough English proofreading and editing.Clarity and problematization in the introduction section needed.Review of literature should cover all the aspects of the topic.Theoretical background must be in line with the topic.Methodology section needs clarity and relevance in each steps. Primary data must be presented properly in results section while the discussion part should be based on the discussion generated on the basis of the key finings. The primary data is somehow underrepresented in the results and discussion parts which need to be properly addressed accordingly. 

We look forward to receiving your revised manuscript.

Kind regards,

Ikram Shah, Ph.D.

Academic Editor

PLOS ONE

Journal Requirements:

5. We note you have included a table to which you do not refer in the text of your manuscript. Please ensure that you refer to Table 3 in your text; if accepted, production will need this reference to link the reader to the Table.

Reviewers' comments:

Reviewer's Responses to Questions

**Comments to the Author**

1. Is the manuscript technically sound, and do the data support the conclusions?

Reviewer #1: Yes

Reviewer #2: Yes

2. Has the statistical analysis been performed appropriately and rigorously? 

Reviewer #1: Yes

Reviewer #2: Yes

3. Have the authors made all data underlying the findings in their manuscript fully available?

Reviewer #1: Yes

Reviewer #2: Yes

4. Is the manuscript presented in an intelligible fashion and written in standard English?

Reviewer #1: Yes

Reviewer #2: Yes

5. Review Comments to the Author

Reviewer #1: Comments to Authors

The article is good contribution to understand the parental relation and activities with performance of their children. However, to make the article more clear to readers, some minor comments are hereby added.

Introduction: introduction is weak in term of discussion of problem in the study area even the literature on national and regional is not well presented.

The objectives and aim of the study should be made part of the introduction instead putting in separate section.

Repetition in introduction, literature and discussion is found in number of places, this should be avoided.

Theory section is limited to only one modal of Bronfenbrenner, other teaching, opportunities and psychological modals have been ignored.

Methodology section:

- The Cross sectional design of the study is followed but the author did not justify why it is necessary for this study?

- The context of multi stage sampling may not be relevant, only saying “District Malakand and its two Tehsils were purposively selected at first and second stage” may not justify the multi stage sampling technique. Purposive sampling with specific criteria would be better suited to such studies.

- Characteristics of respondents and conceptual framework with such headings and short explanation may be avoided and instead may be added in the research design.

Results and discussion:

The major issues I found is that either this article is review article or based on primary data. The discussion is totally taken from secondary sources and primary data has very weak presentation. My suggestions to authors to focus more on your own data, however, you may support the argument with other literature.

The sample selected composed bot boys and girls, similarly government and private schools, but there is no discussion of how girls performed in relation to authoritative, authoritarian and submissive role?? The culture and condition for girls in the study area is very different and difficult, similarly the parents relations to boys and girls in term of education achievement is totally different but I can’t see such discussion or data.

There is also difference in government and private school students performance and attitude of parents towards children studying in both category of schools but such context have been totally ignored.

The recommendations added are beyond the findings, the author should avoid such recommendation and limit these as per result of the research. Recommendation should be revised.

Proofreading, clarity of sentences, abbreviation and repetition need to be corrected and followed in revised submission.

Reviewer #2: The study conducted is overall sound and contains quite a bit of information regarding the topic and findings. The findings of the paper are based on the data presented and the discussion is presented in detail, with adequate reference to other similar studies. In terms of the writing style and grammar of the main body of text, there are some clarity and grammatical issues which must be addressed before the acceptance of the paper. More specific comments on the different sections and aspects are highlighted as follows.

The quality of writing, grammar and use of proper standard English are lacking throughout the manuscript. For example in the sentence "However, certain social factors are much detrimental to put youth at the disadvantaged side in an unequal society, as the benefits of some interventions trickle differently to youth from different socioeconomic groups." many grammatical mistakes are apparent which effect the overall quality of the paper. The authors are strongly recommended to proofread and edit the manuscript for language and proper use of grammar. Personally I feel like the 'how much' in your title "Academic performance of secondary school students in relation to parenting styles and family socioeconomic status: What attributes are important and how much?" does not sound right and could perhaps be reworded for a more suiting phrase.

I feel that the literature on 'parenting styles' especially is not very broad. In the introduction, only one major perspective from literature is repeatedly referred to, indicating a limited and very restrictive conception of the concept and its importance as a variable linked to academic performance. furthermore, there is NO literature included on the concept in the literature review section. Why is this so? the literature review almost exclusively focuses on the relation of socio-economic status with academic performance. The literature review needs to be more balanced, covering the various aspects of the study and the relevant literature thereof.

Regarding the theoretical section of the paper, many aspects of the theory chosen will not be used at all. Why is this? I did not see any justification of why only two layers were used instead of the four presented in the theory.

It would also be nice to include the different aspects of your main variables and how they affect academic outcomes in the introduction. Additionally, why has such a narrow conception of academic performance in the case of this study been used? Academic performance can be based on many variables as reported in the literature, so why does this study ignore all and only focus on one? There is some need for justifications in this regards. Furthermore, although the introduction does introduce the main aspects and topics of the study, the 'probelm' of the study is not adequately defined.

Some of the information given in the first few sections of the paper seem to be over generalized, which should be avoided. For example the statement "The reason for the poor academic achievements of students in the poor and developing countries (South Asian and African) are almost the same". In my opinion such gross generalizations may not be appropriate.

Relating to the following paragraph .......“The domain of occupation/major income source was measured on four levels scale (1=private job/business, 2=agriculture, 3=remittances and 4=government job). Based on the scores level, the highest possible score of the three mentioned domains measuring socioeconomic status (SES) of the family became 14” Thus respondents with a score of 7 or below on the socioeconomic status scale were ranked into Low Socioeconomic status families and those scoring above 7 on the scale were considered as from High Socioeconomic status families"........the classification of occupation or sources of income is not clear to me. Why are private jobs and businesses ranked much lower than Government Jobs and remittances? Is this strictly based on economic terms? Arguably some private jobs and businesses would clearly be more economically lucrative than government jobs, or is this assumption wrong. It would be therefore helpful to discuss why the criteria was set as such, and what the reasons were behind such a ranking.

Although the discussion section is very well linked to current research, this perhaps means less reflection and opinion from the researchers themselves, which would add some richness to this section.

6. PLOS authors have the option to publish the peer review history of their article (what does this mean?). If published, this will include your full peer review and any attached files.

Reviewer #1: No

Reviewer #2: No

---

## [Author Response · Author response to Decision Letter 0]

21 Feb 2023

Review Comments to the Author

Reviewer #1: Comments to Authors

The article is good contribution to understand the parental relation and activities with performance of their children. However, to make the article more clear to readers, some minor comments are hereby added.

Introduction: introduction is weak in term of discussion of problem in the study area even the literature on national and regional is not well presented.

The objectives and aim of the study should be made part of the introduction instead putting in separate section.

Response to reviewer comments 

A comprehensive discussion of problem is included and also literature on national and regional was included in the revised draft

Objectives of the study is added in the revised version in the introduction as suggested 

Repetition in introduction, literature and discussion is found in number of places, this should be avoided.

Theory section is limited to only one modal of Bronfenbrenner, other teaching, opportunities and psychological modals have been ignored.

Response to reviewer

The repeated data is excluded and also different theories (achievement goal theory, self-determination theory, social learning theory, and socio-ecological model) are included. 

Methodology section:

- The Cross sectional design of the study is followed but the author did not justify why it is necessary for this study?

- The context of multi stage sampling may not be relevant, only saying “District Malakand and its two Tehsils were purposively selected at first and second stage” may not justify the multi stage sampling technique. Purposive sampling with specific criteria would be better suited to such studies.

- Characteristics of respondents and conceptual framework with such headings and short explanation may be avoided and instead may be added in the research design.

Response to reviewer

Cross sectional design of the study is followed but the author did not justify why it is necessary for this study

A cross sectional study was adopted to collect data from students of different areas, background so that to compare the results with respect to student gender and their socioeconomic status

Multistage stratified method was replaced by purposive method was adopted for selection of the study area as suggested. 

Separate headings for characteristics of the respondents and conceptual framework are excluded in the revised version. 

Results and discussion:

The major issues I found is that either this article is review article or based on primary data. The discussion is totally taken from secondary sources and primary data has very weak presentation. My suggestions to authors to focus more on your own data, however, you may support the argument with other literature.

A discussion is made over the primary results in the discussion section and the supported literature was also added. 

The sample selected composed bot boys and girls, similarly government and private schools, but there is no discussion of how girls performed in relation to authoritative, authoritarian and submissive role?? The culture and condition for girls in the study area is very different and difficult, similarly the parents relations to boys and girls in term of education achievement is totally different but I can’t see such discussion or data.

Gender based variable is added in the results section also discussion is made on gender based in the discussion section in the revised manuscript, however, family type is not focus in the current study. 

There is also difference in government and private school students performance and attitude of parents towards children studying in both category of schools but such context have been totally ignored.

The current study only focused on the influence of students’ gender, parenting style and family’s socioeconomic status on the academic performance of secondary school students. 

The recommendations added are beyond the findings, the author should avoid such recommendation and limit these as per result of the research. Recommendation should be revised.

The recommendations are revised and align with the study findings. 

Proofreading, clarity of sentences, abbreviation and repetition need to be corrected and followed in revised submission.

The manuscript was proofreaded and the repetitions are avoided. 

Reviewer #2: The study conducted is overall sound and contains quite a bit of information regarding the topic and findings. The findings of the paper are based on the data presented and the discussion is presented in detail, with adequate reference to other similar studies. In terms of the writing style and grammar of the main body of text, there are some clarity and grammatical issues which must be addressed before the acceptance of the paper. More specific comments on the different sections and aspects are highlighted as follows.

The grammatical mistakes where find in the manuscript is corrected in the revised draft

The quality of writing, grammar and use of proper standard English are lacking throughout the manuscript. For example in the sentence "However, certain social factors are much detrimental to put youth at the disadvantaged side in an unequal society, as the benefits of some interventions trickle differently to youth from different socioeconomic groups." many grammatical mistakes are apparent which effect the overall quality of the paper. The authors are strongly recommended to proofread and edit the manuscript for language and proper use of grammar. Personally I feel like the 'how much' in your title "Academic performance of secondary school students in relation to parenting styles and family socioeconomic status: What attributes are important and how much?" does not sound right and could perhaps be reworded for a more suiting phrase.

The title of the manuscript is rephrased 

I feel that the literature on 'parenting styles' especially is not very broad. In the introduction, only one major perspective from literature is repeatedly referred to, indicating a limited and very restrictive conception of the concept and its importance as a variable linked to academic performance. furthermore, there is NO literature included on the concept in the literature review section. Why is this so? the literature review almost exclusively focuses on the relation of socio-economic status with academic performance. The literature review needs to be more balanced, covering the various aspects of the study and the relevant literature thereof.

literature was added on different concept of the article (regional scenario, socioeconomic status, gender, academic performance)

Regarding the theoretical section of the paper, many aspects of the theory chosen will not be used at all. Why is this? I did not see any justification of why only two layers were used instead of the four presented in the theory.

The two level of Bronfenbrenner’s theory (Micro and Meso level) was adopted as these two level directly effect child development, however, the other two levels has indirect effect of children. 

It would also be nice to include the different aspects of your main variables and how they affect academic outcomes in the introduction. Additionally, why has such a narrow conception of academic performance in the case of this study been used? Academic performance can be based on many variables as reported in the literature, so why does this study ignore all and only focus on one? There is some need for justifications in this regards. Furthermore, although the introduction does introduce the main aspects and topics of the study, the 'probelm' of the study is not adequately defined.

In the revised draft the problem statement is discussed in detail, and as the current study was limited only to associate student academic performance with student gender, socioeconomic status and parenting styles only. 

Some of the information given in the first few sections of the paper seem to be over generalized, which should be avoided. For example the statement "The reason for the poor academic achievements of students in the poor and developing countries (South Asian and African) are almost the same". In my opinion such gross generalizations may not be appropriate.

The statements are revised in the revised draft to make it specific 

Relating to the following paragraph .......“The domain of occupation/major income source was measured on four levels scale (1=private job/business, 2=agriculture, 3=remittances and 4=government job). Based on the scores level, the highest possible score of the three mentioned domains measuring socioeconomic status (SES) of the family became 14” Thus respondents with a score of 7 or below on the socioeconomic status scale were ranked into Low Socioeconomic status families and those scoring above 7 on the scale were considered as from High Socioeconomic status families"........the classification of occupation or sources of income is not clear to me. Why are private jobs and businesses ranked much lower than Government Jobs and remittances? Is this strictly based on economic terms? Arguably some private jobs and businesses would clearly be more economically lucrative than government jobs, or is this assumption wrong. It would be therefore helpful to discuss why the criteria was set as such, and what the reasons were behind such a ranking.

In the study area (Pakistan) the government jobs are considered more prestigious and valuable than private jobs and businesses, therefore, the government jobs was considered higher in rank than other occupations. 

Although the discussion section is very well linked to current research, this perhaps means less reflection and opinion from the researchers themselves, which would add some richness to this section.

The discussion section was revised by adding interpretation of the research and also relevant literature was added.

---

## [Decision Letter · Decision Letter 1]

28 Mar 2023

PONE-D-22-24697R1Academic performance of secondary school students in relation to parenting styles, student gender and family socioeconomic status: What attributes are important and what is the extent of their influence?PLOS ONE

Dear Dr. Ali,

Thank you for submitting your manuscript to PLOS ONE. After careful consideration, we feel that it has merit but does not fully meet PLOS ONE’s publication criteria as it currently stands. Therefore, we invite you to submit a revised version of the manuscript that addresses the points raised during the review process. You need to pay significant attention to the following comments raised by one of our esteemed reviewer during the review process. The texts highlighted in yellow color must be incorporated otherwise your paper will not be processed further.   Revised comments

The authors have revised most of the comments, and agree to their response, however, some comments need to be incorporated. The article may be then processed for publication.

I personally not satisfied with title; it does not look professional. This need to be revised and should be between 12-16 words.

My suggestion is to avoid headings in abstract and this should be whole one paragraph, unless not required to journal. Also revised the methodology section in the abstract because you have already revised the sampling techniques. The language in result section of abstract may be corrected, e.g. see the sentence. “The family socioeconomic status emerged as the strongest predictor of the academic performance of children (β=18.25) followed by the interaction of socioeconomic status and authoritative parenting (β=14.18), authoritative parenting (β=13.38), the interaction of socioeconomic status and permissive parenting (β=11.46), and permissive parenting (β=9.2).

The authors claim their response that introduction has been revised as per earlier comments, but I can’t see the comments incorporated. It is good that you reduced the introduction, but you did not introduce the aim and objectives of the study in introduction. The readers would be interested to quickly grasp the main purpose of research in the very beginning.

The section on “National and International Scenario “ may come before the theoretical debate and may be continued with introduction and this would then clarify my above comments on introduction.

The socioeconomic status and monthly income may be revised “(1=monthly income

less than PRs. 15, 000, 2=monthly income PRs. 15,000-30,000 and 3= monthly income above PRs.

30,000)“. Under international standard all the above categories fall below the poverty line and low socio-economic status. This may be inline with minimum wage labour PKRS, 25000, i.e. upto 25000, 26000 to 50000 and 51000 and above.

In conclusion, the second part of this sentences does not clarify the purpose that you are conveying “In addition, the family socioeconomic status emerged as the strongest predictor of the academic performance of children, followed by the interaction of socioeconomic status and authoritative parenting, the interaction of socioeconomic status and permissive parenting, authoritative parenting and permissive parenting” what you want to convey from this sentence??

Under recommendation you have added the following

“The findings also call for the mobilization of charitable resources like Zakat (religious binding on rich Muslims to distribute 2.5% of their financial resources like gold, silver and /or money among needy poor) and Ushr (religious binding on Muslim farmers to distribute 1/10th to 1/20th of their agriculture produce among needy poor) with a strong religious endorsement for supporting educational and other financial needs of families with low socioeconomic status” .

How you relate this with your result and findings? Is there any such findings or analysis?? You may put recommendation only in relation to your variables.

We look forward to receiving your revised manuscript.

Kind regards,

Ikram Shah, Ph.D.

Academic Editor

PLOS ONE

Journal Requirements:

Reviewers' comments:

Reviewer's Responses to Questions

**Comments to the Author**

1. If the authors have adequately addressed your comments raised in a previous round of review and you feel that this manuscript is now acceptable for publication, you may indicate that here to bypass the “Comments to the Author” section, enter your conflict of interest statement in the “Confidential to Editor” section, and submit your "Accept" recommendation.

Reviewer #1: All comments have been addressed

Reviewer #2: All comments have been addressed

2. Is the manuscript technically sound, and do the data support the conclusions?

Reviewer #1: Yes

Reviewer #2: Yes

3. Has the statistical analysis been performed appropriately and rigorously? 

Reviewer #1: Yes

Reviewer #2: Yes

4. Have the authors made all data underlying the findings in their manuscript fully available?

Reviewer #1: Yes

Reviewer #2: Yes

5. Is the manuscript presented in an intelligible fashion and written in standard English?

Reviewer #1: Yes

Reviewer #2: Yes

6. Review Comments to the Author

Reviewer #1: Revised comments

The authors have revised most of the comments, and agree to their response, however, some comments need to be incorporated. The article may be then processed for publication.

I personally not satisfied with title; it does not look professional. This need to be revised and should be between 12-16 words.

My suggestion is to avoid headings in abstract and this should be whole one paragraph, unless not required to journal. Also revised the methodology section in the abstract because you have already revised the sampling techniques. The language in result section of abstract may be corrected, e.g. see the sentence. “The family socioeconomic status emerged as the strongest predictor of the academic performance of children (β=18.25) followed by the interaction of socioeconomic status and authoritative parenting (β=14.18), authoritative parenting (β=13.38), the interaction of socioeconomic status and permissive parenting (β=11.46), and permissive parenting (β=9.2).

The authors claim their response that introduction has been revised as per earlier comments, but I can’t see the comments incorporated. It is good that you reduced the introduction, but you did not introduce the aim and objectives of the study in introduction. The readers would be interested to quickly grasp the main purpose of research in the very beginning.

The section on “National and International Scenario “ may come before the theoretical debate and may be continued with introduction and this would then clarify my above comments on introduction.

The socioeconomic status and monthly income may be revised “(1=monthly income

less than PRs. 15, 000, 2=monthly income PRs. 15,000-30,000 and 3= monthly income above PRs.

30,000)“. Under international standard all the above categories fall below the poverty line and low socio-economic status. This may be inline with minimum wage labour PKRS, 25000, i.e. upto 25000, 26000 to 50000 and 51000 and above.

In conclusion, the second part of this sentences does not clarify the purpose that you are conveying “In addition, the family socioeconomic status emerged as the strongest predictor of the academic performance of children, followed by the interaction of socioeconomic status and authoritative parenting, the interaction of socioeconomic status and permissive parenting, authoritative parenting and permissive parenting” what you want to convey from this sentence??

Under recommendation you have added the following

“The findings also call for the mobilization of charitable resources like Zakat (religious binding on rich Muslims to distribute 2.5% of their financial resources like gold, silver and /or money among needy poor) and Ushr (religious binding on Muslim farmers to distribute 1/10th to 1/20th of their agriculture produce among needy poor) with a strong religious endorsement for supporting educational and other financial needs of families with low socioeconomic status” .

How you relate this with your result and findings? Is there any such findings or analysis?? You may put recommendation only in relation to your variables.

Reviewer #2: I am satisfied with the changes made by the authors in response to my comments. Almost all issues were dealt with, which has improved the quality of the manuscript and therefore I now recommend it for publication.

7. PLOS authors have the option to publish the peer review history of their article (what does this mean?). If published, this will include your full peer review and any attached files.

Reviewer #1: No

Reviewer #2: No

---

## [Author Response · Author response to Decision Letter 1]

7 Apr 2023

Revised comments

1. I personally not satisfied with title; it does not look professional. This need to be revised and should be between 12-16 words.

Response: the title is revised accordingly to suggested range of words. 

2. My suggestion is to avoid headings in abstract and this should be whole one paragraph, unless not required to journal. 

Response: abstract is revised as suggested 

3. Also revised the methodology section in the abstract because you have already revised the sampling techniques. 

Response: methodology section in the abstract is revised as suggested and aligned with the sampling technique used. 

4. The language in result section of abstract may be corrected, e.g. see the sentence. “The family socioeconomic status emerged as the strongest predictor of the academic performance of children (β=18.25) followed by the interaction of socioeconomic status and authoritative parenting (β=14.18), authoritative parenting (β=13.38), the interaction of socioeconomic status and permissive parenting (β=11.46), and permissive parenting (β=9.2).

Response: the paragraph iss revised while correcting the language. 

5. The authors claim their response that introduction has been revised as per earlier comments, but I can’t see the comments incorporated. It is good that you reduced the introduction, but you did not introduce the aim and objectives of the study in introduction. The readers would be interested to quickly grasp the main purpose of research in the very beginning.

Response: aim and objectives of the study is included in the revised version

6. The section on “National and International Scenario “ may come before the theoretical debate and may be continued with introduction and this would then clarify my above comments on introduction.

Response: rearranged this section as suggested

7. The socioeconomic status and monthly income may be revised “(1=monthly income

less than PRs. 15, 000, 2=monthly income PRs. 15,000-30,000 and 3= monthly income above PRs.

30,000)“. Under international standard all the above categories fall below the poverty line and low socio-economic status. This may be inline with minimum wage labour PKRS, 25000, i.e. upto 25000, 26000 to 50000 and 51000 and above.

Response: the attributes of SES is aligned with minimum wage labour as suggested: PKRS, 25000, i.e. up to 25000, 26000 to 50000 and 51000 and above.

8. In conclusion, the second part of this sentences does not clarify the purpose that you are conveying “In addition, the family socioeconomic status emerged as the strongest predictor of the academic performance of children, followed by the interaction of socioeconomic status and authoritative parenting, the interaction of socioeconomic status and permissive parenting, authoritative parenting and permissive parenting” what you want to convey from this sentence??

Response: the sentence is explained and revised to make it clearer

9. Under recommendation you have added the following

“The findings also call for the mobilization of charitable resources like Zakat (religious binding on rich Muslims to distribute 2.5% of their financial resources like gold, silver and /or money among needy poor) and Ushr (religious binding on Muslim farmers to distribute 1/10th to 1/20th of their agriculture produce among needy poor) with a strong religious endorsement for supporting educational and other financial needs of families with low socioeconomic status” . How you relate this with your result and findings? Is there any such findings or analysis?? You may put recommendation only in relation to your variables.

Response: the recommendation was revised and related with the study findings on SES and academic performance.

---

## [Editor Report · Decision Letter 2]

13 Apr 2023

PONE-D-22-24697R2Academic performance of children in relation to gender, parenting styles, and socioeconomic status: what attributes are importantPLOS ONE

Dear Dr. Nayab Ali,

Thank you for submitting your revised manuscript to PLOS ONE. After careful consideration, we feel that it has merit but does not fully meet PLOS ONE’s publication criteria as it currently stands. Therefore, we invite you to submit a revised version with minor changes of your manuscript that addresses the points raised during the review process. 1. Rephrase the sentence in introduction section "*Also, when the state has more citizens with poor academic skills, they are not able to introduce productivity-enhancing technologies and new......*"  2. Avoid the abbreviations such as etc3. The sentence  starting from "These factors include individual, social and institutional characteristics and so on" the phrase "so on" does not make any sense you have to explain all the factors or factors of your interest here in detail, the phrase "so on" makes the sentence ambiguous.4. Aim and objectives should be presented in introduction section without heading and subheading(s). For this purpose a dedicated single paragraph may be used to present them. 5. In methodology section, details on the study area must be added for the readers who are not familiar with local context and arrangements.6. Rephrase the sentence of Cronbach value of the scale used in this study that should indicate the value of each scale for this study within the accepted range.7. Although it is well explained in the ethical statement that informed consent was taken from Head of the educational institute but little has been mentioned that how informed consent was taken from students.  8. A more detail may be added for the protocol of each interview. 9. Conflict of interest must be presented just after the conclusion. 10. The asterisk in Table 7 must be defined in parenthesis.11. In discussion some sentences are quite interesting and confusing such as "Thus the democratic culture promotes............, The author added................., Researcher himself (which indicates that a single research involvement in this study). Check all these sentences for correction and relevance with your study. 12. In conclusion the authors claim that this study has policy implications but could not provide any detail on how and where aspects of policy implication. 13. The author need to argue that why gender variable is not significant in this study with scientific reason(s).14. Few practical recommendations may be present that should be within the mandate of legal and policy instruments. Irrelevant recommendation may be removed.   15. Please check reference number 2, 8, 13, 19, and 25.  

We look forward to receiving your revised manuscript.

Kind regards,

Ikram Shah, Ph.D.

Academic Editor

PLOS ONE
---

## [Author Response · Author response to Decision Letter 2]

27 Apr 2023

Response to editor comments 

1. Rephrase the sentence in introduction section "Also, when the state has more citizens with poor academic skills, they are not able to introduce productivity-enhancing technologies and new......" 

Response: rephrased the line in the revised version

2. Avoid the abbreviations such as etc

Response: the abbreviations are elaborated in the revised version

3. The sentence starting from "These factors include individual, social and institutional characteristics and so on" the phrase "so on" does not make any sense you have to explain all the factors or factors of your interest here in detail, the phrase "so on" makes the sentence ambiguous.

Response: corrected in the revised version

4. Aim and objectives should be presented in introduction section without heading and subheading(s). For this purpose a dedicated single paragraph may be used to present them. 

Response: corrected as suggested

5. In methodology section, details on the study area must be added for the readers who are not familiar with local context and arrangements.

Response: more detail about the study area is provided in the revised version

6. Rephrase the sentence of Cronbach value of the scale used in this study that should indicate the value of each scale for this study within the accepted range.

Response: Cronbach value of each scale was added 

7. Although it is well explained in the ethical statement that informed consent was taken from Head of the educational institute but little has been mentioned that how informed consent was taken from students. 

Response: Cronbach value of each scale was added 

8. A more detail may be added for the protocol of each interview. 

Response: protocol of each interview was explained in the revised draft 

9. Conflict of interest must be presented just after the conclusion. 

Response: placed as suggested 

10. The asterisk in Table 7 must be defined in parenthesis.

Response: the asterisk in Table 7 are defined in parenthesis

11. In discussion some sentences are quite interesting and confusing such as "Thus the democratic culture promotes............, The author added................., Researcher himself (which indicates that a single research involvement in this study). Check all these sentences for correction and relevance with your study. 

Response: these sentences were revised and make it in-line with the study 

12. In conclusion the authors claim that this study has policy implications but could not provide any detail on how and where aspects of policy implication. 

Response: the policy implication is well highlighted in the revised draft 

13. The author needs to argue that why gender variable is not significant in this study with scientific reason(s).

Response: the explanation for this is provided in the revised draft

14. Few practical recommendations may be present that should be within the mandate of legal and policy instruments. Irrelevant recommendation may be removed. 

Response: Policy implications that were highlighted in the conclusions are also clubbed in the recommendations therefore, the recommendations is explained to make it aline with the conclusion and I think there is no irrelevant recommendation included 

15. Please check reference number 2, 8, 13, 19, and 25. 

Response: corrected

---

## [Editor Report · Decision Letter 3]

24 May 2023

Academic performance of children in relation to gender, parenting styles, and socioeconomic status: what attributes are important

PONE-D-22-24697R3

Dear Dr. Nayab Ali,

We’re pleased to inform you that your manuscript has been judged scientifically suitable for publication and will be formally accepted for publication once it meets all outstanding technical requirements.

Kind regards,

Ikram Shah, Ph.D.

Academic Editor

PLOS ONE
---

## [Editor Report · Acceptance letter]

26 Jun 2023

PONE-D-22-24697R3 

Academic performance of children in relation to gender, parenting styles, and socioeconomic status: What attributes are important 

Dear Dr. Ali:

I'm pleased to inform you that your manuscript has been deemed suitable for publication in PLOS ONE. Congratulations! Your manuscript is now with our production department. 

Kind regards, 

on behalf of

Dr. Ikram Shah 

Academic Editor

PLOS ONE